Mapping ISO 19115-1 geographic metadata standards to CodeMeta

Habermann Ted ted.habermann@gmail.com
The HDF Group , Champaign , IL , United States of America
Peroni Silvio
Electronic publication date: 2019 Feb 4
Publication date: 2019
Volume: 5
Electronic Location ID: e174
Received 2018 Aug 28; Accepted 2019 Jan 11
Copyright: ©2019 Habermann
Copyright year: 2019
Copyright holder: Habermann
License: This is an open access article distributed under the terms of the Creative Commons Attribution License, which permits unrestricted use, distribution, reproduction and adaptation in any medium and for any purpose provided that it is properly attributed. For attribution, the original author(s), title, publication source (PeerJ Computer Science) and either DOI or URL of the article must be cited.
License URL: https://creativecommons.org/licenses/by/4.0/

Keywords: Software citation, CodeMeta, ISO metadata, Metadata, Crosswalk

Funding: National Science Foundation NSFDACS11C1675 NASA/GSFC NNG15HZ39C All of this material is based upon work supported by the National Science Foundation under Grant No. NSFDACS11C1675 and by NASA/GSFC under Raytheon Co. contract number NNG15HZ39C. There was no additional external funding received for this study. The funders had no role in study design, data collection and analysis, decision to publish, or preparation of the manuscript.

==============================
The CodeMeta Project recently proposed a vocabulary for software metadata. ISO Technical Committee 211 has published a set of metadata standards for geographic data and many kinds of related resources, including software. In order for ISO metadata creators and users to take advantage of the CodeMeta recommendations, a mapping from ISO elements to the CodeMeta vocabulary must exist. This mapping is complicated by differences in the approaches used by ISO and CodeMeta, primarily a difference between hard and soft typing of metadata elements. These differences are described in detail and a mapping is proposed that includes sixty-four of the sixty-eight CodeMeta V2 terms. The CodeMeta terms have also been mapped to dialects used by twenty-one software repositories, registries and archives. The average number of terms mapped in these cases is 11.2. The disparity between these numbers reflects the fact that many of the dialects that have been mapped to CodeMeta are focused on citation or dependency identification and management while ISO and CodeMeta share additional targets that include access, use, and understanding. Addressing this broader set of use cases requires more metadata elements.

Introduction

The CodeMeta Project (CodeMeta Project, 2018) recently proposed (1) a vocabulary for documenting software, (2) mappings between metadata fields used by a broad range of software repositories, registries and archives (CodeMeta Crosswalk, 2018), and (3) developed software with the purpose of facilitating automatic translation between different representations of software metadata. The vocabulary was designed to support several different software use cases, including citation, discovery, use and, to some degree, understanding.

The ISO Technical Committee 211 has developed generic metadata standards that are widely used for geographic data of many kinds. These standards were also designed as a foundation that can be built on to document many kinds of things and support many use cases (Habermann, 2018a; Habermann, 2018b; Habermann, 2018c).

This paper describes a mapping between the conceptual model that underlies ISO metadata (ISO 19115-1, 2014) and CodeMeta with the goal of facilitating the creation of CodeMeta-compliant descriptions of software that is documented using the ISO standards. The communities that developed these two metadata dialects share the important goal of comprehensive standards that address multiple use cases for many disciplines. Both groups pursue this goal by developing consensus, but the details of the processes used to develop their standards differ. ISO TC211 represents a traditional International standards body with well-defined processes, publication methods and a business model that includes costs to users for standards documents. CodeMeta represents a community of volunteer practitioners with an initial set of proposed conventions open on the Web and an invitation for adoption, experimentation and evolution.

In addition to these process differences, there are also differences between the structures and implementations of these two models. These are described below with the mappings following.

Dialect coverage and scope

Mapping metadata for software between different schemas and dialects is an important technical goal of CodeMeta. This goal is supported using a crosswalk file that is maintained and contributed to in the CodeMeta Git repository (CodeMeta Crosswalk, 2018). This file lists the CodeMeta terms along with equivalents in twenty-one dialects. This crosswalk is the basis for translating content between these dialects.

A similar situation occurs in many in science communities that are trying to support multiple use cases, i.e., document, share, and trust, for datasets using multiple metadata dialects (see Gordon & Habermann, 2018). The concept of “Dialect Coverage” has come up in those studies as the amount (%) of the concepts in a particular recommendation that a dialect includes. In the CodeMeta case, this is the number of CodeMeta concepts that can be represented in the dialects listed in the crosswalk file (Crosswalk Data, 2018). Figure 1 shows this count for each of the twenty-one dialects. These counts were determined from the crosswalk file by counting the number of cells with content in each column using a spreadsheet count() function. Both versions of CodeMeta and ISO 19115-1 (2016) are included in this figure as well.

Figure 1 Coverage of CodeMeta concepts in multiple dialects.

The average number of CodeMeta concepts covered by twenty-one dialects is 11.2. The ISO dialect covers sixty-four of sixty-eight CodeMeta concepts.

The data show that the ISO dialect covers very close to all (64/68) of the CodeMeta concepts and the other twenty-one dialects cover an average of 11.23/68, suggesting that CodeMeta is more similar to ISO than it is to other dialects that have done crosswalks. The difference between the ISO mapping and others is striking. It likely reflects the difference between the small number of metadata elements used for discovering and citing software (or data) and the larger number needed to be able to use it and trust it. In the current software citation landscape, this is the difference between the complete CodeMeta vocabulary, i.e., all metadata for code (over sixty items), and the FORCE11 Software Citation Guidelines, i.e., metadata for code citation (Smith, Katz & Niemeyer, 2016) which includes only ten items.

In some cases, multiple CodeMeta terms are mapped to the same ISO elements. Most of this ambiguity is related to differences in the two models and approaches that are discussed in detail below. ISO elements that are mapped to more than one CodeMeta term are identified with * in the crosswalk tables below.

Model characteristics

The ISO metadata standards are based on a UML model that is harmonized across all standards developed and managed by the committee. The model is built around classes and attributes that describe the structure of the standards and the relationships among objects. ISO 19115-1 (2014) includes thirteen top-level classes that provide details on identification, content, constraints, distribution, quality, usage, reference systems, spatial representation and several other areas.

The ISO standard includes a scope element at the root of each record that gives the type of resource described by the metadata. The default scope is dataset, but other options include: aggregate, application, attribute, attributeType, collection, collectionHardware, collectionSession, coverage, dimensionGroup, document, feature, featureType, fieldSession, initiative, metadata, model, nonGeographicDataset, otherAggregate, platformSeries, product, productionSeries, propertyType, repository, sample, sensor, sensorSeries, series, service, software, tile, transferAggregate (see Habermann, 2018a; Habermann, 2018b; Habermann, 2018c). Mapping the CodeMeta vocabulary to the ISO standard is an initial step toward defining the content that could be included in ISO metadata records that describe software and applications, i.e., those where the scope is software.

The most commonly used representation of the ISO standards is XML (ISO 19115-1, 2016). ISO XPaths uniquely identify metadata content and follow the structure of the UML model, with levels in the XML alternating between objects (with types) and properties. This results in XML that is “striped” like the XML representation of the Resource Description Framework (RDF, 2018) (W3C, 2014), i.e., role/type/role/type/content. Types generally start with two uppercase letters (MD, CI, …) that indicate the UML package that they are defined in (metadata, citation, …) followed by an underscore (MD_, CI_, …). Properties (termed roles in this discussion) are in lower camel case.

A significant benefit of the striped XML is that properties can be defined with abstract objects that can share properties while being instantiated with different types. For example, the ISO CI_Party object is abstract and includes name and contactInfo properties. It is extended and specialized by CI_Individual and CI_ Organisation objects which inherit name and contactInfo properties and add properties that are relevant for people and organizations, e.g., organizations can include individuals, logos, and position names. This approach also facilitates reuse by allowing standard objects (e.g., people, organizations, or citations) to be referenced using links rather than repetitive content (https://geo-ide.noaa.gov/wiki/index.php?title=ISO_Components).

Another benefit of this approach in ISO is the same as that in the schema.org case—communities can extend object definitions when necessary and, in the ISO case, the resulting extended objects fit naturally into the ISO XML representation. This approach is similar to the schema extension model used in CodeMeta to add properties deemed important by the CodeMeta community to the more general SoftwareSourceCode schema that is also a specialization of the schema.CreativeWork schema.

The namespace for each element in the XML is identified using a standard namespace prefix (mdb, cit, …). Asterisks are used in the XPaths to indicate locations where several objects can be used. For example, mdb:identificationInfo/*/ indicates that either mdb:MD_ DataIdentification or srv:SV_ServiceIdentification objects can occur in that location.

A simplified notation is introduced for paths through the UML conceptual model in this document that includes only the role names and no information that is specific to the XML representation. For example, the XPath /mdb:MD_Metadata/mdb:identificationInfo/ mri:MD_ DataIdentification/mri:resourceSpecificUsage/mri:MD_ Usage/mri:identifiedIssues/ cit:CI_ Citation/cit:onlineResource/cit:CI_OnlineResource/cit:linkage is replaced by the concept path: identificationInfo.resourceSpecificUsage.identifiedIssues.onlineResource.linkage. These simplified “concept paths” improve readability and emphasize equivalences between CodeMeta and ISO in the conceptual space. Specific XPaths can be constructed from these concept paths when necessary to implement translation of existing ISO content to CodeMeta representations. The reverse translation is not unique.

CodeMeta specifies a vocabulary rather than a structural model. It includes properties from several schema.org schemas listed in Table 1 along with the number of items from each schema (Crosswalk Data, 2018). These schemas exist in a schema.org hierarchy which is similar in many ways to the ISO structure. SoftwareApplication and SoftwareSourceCode schemas are both specializations of the Thing >CreativeWork schema. CodeMeta extends these schemas (in CodeMeta.SoftwareSourceCode) with several properties that lack clear equivalents in schema.org.

Table 1 Schema.org schemas and item counts for CodeMeta vocabulary.

Source	#Terms	Source	#Terms	
schema:CreativeWork	24	schema:Thing	6	
schema:SoftwareApplication	15	schema:SoftwareSourceCode	4	
CodeMeta:SoftwareSourceCode	10	schema (not mapped)	2	
schema:Person	7			

Hard types and soft types

All standards and vocabularies need to make choices between hard or soft typing of objects they are describing. Hard typing requires specific names for items and is the only choice available in implementations where names alone can be used to distinguish between items, e.g., CodeMeta. For example, if publication and revision dates are required for complete descriptions of a resource, hard typed representations would include two items: e.g., publicationDate and revisionDate. Soft Typing can be used in dialects which support item attributes as well as values, e.g., XML. In that case, these two dates would be represented with the same name (XML element) and distinguished by a type attribute: <datetype=”publication”>and <datetype=”revision”>.

The difference between these two approaches emerges as the dialects evolve. Hard types evolve by adding new elements to the underlying model, e.g., adding creationDate (or some other type of date) when it becomes apparent that it is needed, and unambiguous definitions of those elements. Soft types evolve by adding items to the shared vocabulary of date types, typically a codelist or thesaurus.

The critical difference between hard and soft types boils down to differences in governance models and change tolerance. In communities that use hard types, members must be tolerant to changes in the models and, typically, changes in tooling built on them. Communities that use soft typing must have mechanisms for sharing and evolving vocabularies, typically control bodies or rules.

The ISO model is soft-typed and the CodeMeta model is hard-typed. Table 2 lists some of the documentation concepts that illustrate the contrast between these approaches. The first row shows the differences in how dates are treated. The CodeMeta vocabulary includes four types of dates listed in the second column of Table 2. If other date types are required to describe software, maybe dateDeprecated for example, new terms would be found in schema.org or added to the vocabulary to address those needs. The ISO approach involves a single date concept and a codelist that includes sixteen options, shown in the third column of Table 2. That codelist is designed to be extended by communities with other needs without impacting the structure of the standard. In this example, the date type codelist already includes the term “deprecated”.

Table 2 CodeMeta hard types and related ISO codelists.

Item	CodeMeta items	ISO 19115-1 codelist valuesa	
Dates	embargoDate dateCreated dateModified datePublished	CI_DateTypeCode: creation publication revision expiry lastUpdate lastRevision nextUpdate unavailable inForce adopted deprecated superseded validityBegins validityExpires released distribution	
People and organizations	author contributor creator copyrightHolder editor funder producer provider publisher sponsor affiliation	CI_RoleCode: resourceProvider custodian owner user distributor originator pointOfContact principalInvestigator processor publisher author sponsor coAuthor collaborator editor mediator rightsHolder contributor funder stakeholder maintainer	
Online resource types	buildInstructions contIntegration issueTracker readme id identifier downloadUrl installUrl codeRepository relatedLink sameAs url	CI_OnLineFunctionCode: download information offlineAccess order search completeMetadata browseGraphic upload emailService browsing fileAccess	
Associations	supportingData	DS_AssociationTypeCode: crossReference largerWorkCitation partOfSeamlessDatabase stereoMate isComposedOf collectiveTitle series dependency revisionOf	
Keyword	Keywords programmingLanguage applicationCategory applicationSubCategory	MD_KeywordTypeCode: discipline place stratum temporal theme dataCentre featureType instrument platform process project service product subTopicCategory taxon	
Notes.

a ISO 19115-1 Codelists from https://standards.iso.org/iso/19115/resources/Codelists/cat/codelists.html.

Citations

Connecting users to resources is one of the most important roles of metadata. It is also one of the most ubiquitous. Several classes of citations are important. Citation to the resource being described in the metadata (Resource Citation). The role of these citations is to provide guidance on how the resource being described should be cited and there is only one of these in each metadata record. Citations to related resources (Related Resource Citation). These are generic references to some other resource and generally include information about the relationship between the resource being described and the related resource. See, for example, the RelatedIdentifier element in the DataCite metadata schema (DataCite Metadata Working Group, 2017) which includes relatedIdentifierType and relationType attributes as additional information. Citations to other, typically specific, resources (Specific Resource Citations). For example, the ISO object that describes data processing includes a citation in the role of softwareReference that specifically provides a reference to software used in the processing.

Other examples of these citation types are included in the following discussion.

ISO citations

ISO 19115-1 includes all three types of citations:

• The Resource Citation is unique and occurs at a specific location in the conceptual model: identificationInfo.citation (XPath = /mdb:MD_Metadata/mdb:identification Info/*/mri:citation/cit:CI_Citation).

• Related Resource Citations also occur at a specific location in the model, identificationInfo.associatedResource (XPath = /mdb:MD_Metadata/mdb:identificationInfo/*/ mri:associatedResource/mri:MD_AssociatedResource/mri:name/cit:CI_Citation along with two codelists (associationType and initiativeType) that provide information about how the resource is associated.).

• Specific Resource Citations occur in a number of locations in the ISO model as part of specific classes. For example, citations to additional documentation occur at identificationInfo.additionalDocumentation and citations to quality reports occur at dataQualityInfo.standaloneQualityReport.

All ISO Citations include elements of traditional citations to books or papers e.g., title, authors (people or organizations in many roles), dates (many types), series information, page numbers, etc., as well as identifiers (ISSN, ISBN, and other types) and URLs with titles, descriptions and types. The XPaths to these items from each ISO citation root are shown in Table 3.

Table 3 Relative xPaths to titles, identifiers, and URLs in ISO citations.

Item	xPath from CI_Citation	
Title	cit:CI_Citation/cit:title/gco:CharacterString (concept = title)	
Identifier	cit:CI_Citation/cit:identifier/mcc:MD_Identifier/mcc:code/gco:CharacterString (concept = identifier.code)	
URL	cit:CI_Citation/cit:onlineResource/cit:CI_OnlineResource/cit:linkage/gco:CharacterString (concept = onlineResource.linkage)	

CodeMeta citations

CodeMeta includes twenty-six terms that represent resources that are related to or support the use of the software being described. These terms have several different types (Text, URL, Text or URL, CreativeWork, CreativeWork or URL, Computer Language or text, …). In the mappings below, these terms are mapped to the ISO citations. The specific types can be described by adding the paths in Table 3 to the concept or XPaths.

Distribution

Many of the distribution systems for geographic data described by ISO metadata include repositories (generally called archives or data centers) that manage and preserve data while providing on-going support for users. ISO metadata standards accommodate approaches to resource distribution with or without descriptions of repositories (termed distributors) and each repository can provide several URLs (transferOptions) for each resource. These onlineResources can have any of the functions included in the CI_OnLineFunctionCode codelist in Table 2. The most common online functions are download and information and these are used in the mappings to indicate direct access to the resource (function =download) or information about the resource (function =information).

Additional documentation

The CodeMeta vocabulary includes many items that are intended to help users use and understand the software described in the metadata. In the ISO standards, these items can be described in two ways: as associated resources (identificationInfo.associatedResource) or as additional documentation (identificationInfo.additionalDocumentation). I have chosen the later in these cases. In dialects without specific citations, e.g., Datacite, these would be referred to as relatedIdentifiers with appropriate relationTypes as the DataCite dialect is soft typed (DataCite Metadata Working Group, 2017).

One important goal of CodeMeta is to enable authors to cite software that is used to store, process, analyze, and visualize the data and model results that they use in their work. Increasing citations from the scientific literature is large part of this goal, but there are also significant opportunities to improve the completeness of dataset metadata by citing software. This is generally done as part of the provenance or lineage section of the metadata. The ISO standards provide several specific resource citations for citing software, including:

• resourceLineage.processStep.processingInformation.algorithm.citation,

• resourceLineage.processStep.processingInformation.softwareReference, and

• resourceLineage.processStep.processingInformation.documentation.

Mappings

The mappings between CodeMeta and ISO are presented here in a series of tables that correspond to the source schema.org schemas used in order to provide some structure that may help clarify the relationships and improve understanding. The process of creating the mappings involved three steps: (1) obvious connections, i.e., description ->identificationInfo.abstract, or name ->identificationInfo.citation.title, (2) more complicated connections like those discussed above, and (3) matching intended types as closely as possible, i.e., CodeMeta terms that were intended to be identifiers were mapped to ISO identifier codes (identifier ->identificationInfo.citation.identifier.code) and those that were intended to be URLs were mapped to ISO linkages (downloadUrl ->distributionInfo.transferOptions.onLine[function =’download’].linkage). This process is, of course, more subjective than objective, and the resulting mappings reflect experience authoring the ISO standards and working with them in multiple contexts over the last decade.

The mappings include the property names, types, and descriptions from the CodeMeta vocabulary, conceptual paths for the ISO items (ISO 19115-1, 2014), and XPaths from the standard XML representation (ISO 19115-1, 2016). The conceptual paths are provided here in lieu of the ISO definitions for simplicity. The complete ISO conceptual model with definitions is available in an HTML view (ISO Conceptual Model, 2018). In some cases, multiple CodeMeta terms are mapped to single ISO elements, as described in the Additional Documentation section above. These cases are marked with * in the tables.

These mappings are also available in machine-readable forms (Habermann, 2018b; Habermann, 2018c).

Schema:Person

The schema.Person schema provides a vocabulary for properties of people. In the ISO standards, people and organizations are both referred to as parties and names can be given as any combination of individual names, organization names, or positions. This mapping includes seven items listed in Table 4.

Table 4 Mapping of CodeMeta terms from the schema.Person schema to ISO 19115-1 and ISO 19115-3.

Property	Type	Description	ISO 19115-1	ISO 19115-3	
address	PostalAddress or Text	Physical address of the item.	party.contactInfo.address. deliveryPoint	cit:party/cit:CI_Organisation/ cit:contactInfo/cit:CI_Contact/ cit:address/cit:CI_Address/cit: deliveryPoint/gco:CharacterString	
affiliation	Text	An organization that this person is affiliated with. For example, a school/university	party.namea	cit:party/cit:CI_Organisation/ cit:name/gco:CharacterString	
email	Text	Email address	party.contactInfo.address. electronicMailAddress	cit:party/cit:CI_Organisation/ cit:contactInfo/cit:CI_Contact/ cit:address/cit:CI_Address/ cit:electronicMailAddress/ gco:CharacterString	
familyName	Text	Family name. In the US the last name of an Person. This can be used along with givenName instead of the name property.	party.namea	cit:party/cit:CI_Individual/ cit:name/gco:CharacterString	
givenName	Text	Given name. In the US the first name of a Person. This can be used along with familyName instead of the name property	party.namea	cit:party/cit:CI_Individual/ cit:name/gco:CharacterString	
identifier	URL	URL identifer, ideally an ORCID ID for individuals, a FundRef ID for funders	party.partyIdentifier.code	cit:party/cit:CI_Organisation/ cit:partyIdentifier/mcc: MD_Identifier/mcc:code/ gco:CharacterString	
name	Text	The name of an Organization, or if separate given and family names cannot be resolved, for a Person	party.namea	cit:party/cit:CI_Organisation/ cit:name/gco:CharacterString	
Notes.

a Multiple CodeMeta terms are mapped to this ISO XML element, some with different attributes.

Schema:Thing

The schema.Thing schema provides a vocabulary for properties of the most generic type of item. In the context of CodeMeta, this item is the resource described by the metadata which is software. In ISO 19115-1 (2014), properties related to the identification of the resource being described are in the identificationInfo section and many of the properties are included in the citation to that resource. As described above, these properties (title, identifier, and link) are included in all citations in the ISO model. This mapping includes six items listed in Table 5.

Table 5 Mapping of CodeMeta terms from the schema.Thing schema to ISO 19115-1 and ISO 19115-3.

Property	Type	Description	ISO 19115-1	ISO 19115-3	
Description	Text	A description of the item.	identificationInfo.abstract	/mdb:MD_Metadata/mdb:identificationInfo/*/mri:abstract/gco:CharacterString	
identifier	PropertyValue or URL	The identifier property represents any kind of identifier for any kind of Thing, such as ISBNs, GTIN codes, UUIDs etc. Schema.org provides dedicated properties for representing many of these, either as textual strings or as URL (URI) links. See background notes for more details.	identificationInfo.citation.identifier.code	/mdb:MD_Metadata/mdb:identificationInfo/*/mri:citation/cit:CI_Citation/cit:identifier/ mcc:MD_Identifier/mcc:code	
name	Text	The name of the item (software, Organization)	identificationInfo.citation.title	/mdb:MD_Metadata/mdb:identificationInfo/*/mri:citation/cit:CI_Citation/cit:title/ gco:CharacterString	
relatedLink	URL	A link related to this object, e.g., related web pages	identificationInfo.citation.onlineResource [function =’information’]a	/mdb:MD_Metadata/mdb:identificationInfo/*/mri:citation/cit:CI_Citation/cit:onlineResource/ cit:CI_OnlineResource/mdb:MD_Metadata/mdb:identificationInfo/*/mri:citation/ cit:CI_Citation/cit:onlineResource/cit:CI_OnlineResource[gmd:function/ gmd:CI_OnLineFunctionCode,’information’]/cit:linkage	
sameAs	URL	URL of a reference Web page that unambiguously indicates the item’s identity. E.g. the URL of the item’s Wikipedia page, Wikidata entry, or official website.	identificationInfo.citation.onlineResource [function =’information’]a	/mdb:MD_Metadata/mdb:identificationInfo/*/mri:citation/cit:CI_Citation/cit:onlineResource/ cit:CI_OnlineResource/mdb:MD_Metadata/mdb:identificationInfo/*/mri:citation/ cit:CI_Citation/cit:onlineResource/cit:CI_OnlineResource[gmd:function/ gmd:CI_OnLineFunctionCode,’information’]/cit:linkage	
url	URL	URL of the item.	identificationInfo.citation.onlineResource [function =’download’]a	/mdb:MD_Metadata/mdb:identificationInfo/*/mri:citation/ cit:CI_Citation/cit:onlineResource/cit:CI_OnlineResource[gmd:function/ gmd:CI_OnLineFunctionCode,’download’]/cit:linkage	
Notes.

a Multiple CodeMeta terms are mapped to this ISO XML element, some with different attributes.

Schema:Thing.CreativeWork

The Thing.CreativeWork schema provides a vocabulary for the most generic kind of creative work, including books, movies, photographs, software programs, etc. This mapping includes twenty-four items listed in Table 6.

Table 6 Mapping of CodeMeta terms from the Thing.CreativeWork schema to ISO 19115-1 and ISO 19115-3.

Property	Type	Description	ISO 19115-1	ISO 19115-3	
aAuthor	Organization or person	The author of this content or rating. Please note that author is special in that HTML 5 provides a special mechanism for indicating authorship via the rel tag. That is equivalent to this and may be used interchangeably.	identificationInfo.citation. citedResponsibleParty[role =’author’].party.name or identificationInfo.citation. citedResponsibleParty[role =’originator’].party.name	/mdb:MD_Metadata/mdb:identificationInfo/*/mri:citation/cit:CI_Citation/ cit:citedResponsibleParty/cit:CI_Responsibility[cit:role/cit:CI_RoleCode =’author’] or /mdb:MD_Metadata/mdb:identificationInfo/*/mri:citation/cit:CI_Citation/ cit:citedResponsibleParty/cit:CI_Responsibility[cit:role/cit:CI_RoleCode =’originator’]	
citation	CreativeWork or URL	A citation or reference to another creative work, such as another publication, web page, scholarly article, etc.	identificationInfo.associatedResource.namea	/mdb:MD_Metadata/mdb:identificationInfo/*/mri:associatedResource/ mri:MD_AssociatedResource/mri:name/cit:CI_Citation	
contributor	Organization or Person	A secondary contributor to the CreativeWork or Event.	identificationInfo.citation.citedResponsibleParty [not(role =’author’ or role =’principalInvestigator’ or role =’originator’)].party.namea	/mdb:MD_Metadata/mdb:identificationInfo/*/mri:citation/ cit:CI_Citation/cit:citedResponsibleParty/cit:CI_Responsibility[not (cit:role/cit:CI_RoleCode =’author’ or cit:role/cit:CI_RoleCode =’principalInvestigator’ or cit:role/cit:CI_RoleCode =’originator’)]/cit:party/*/cit:name	
copyright Holder	Organization or Person	The party holding the legal copyright to the CreativeWork.	identificationInfo.resourceConstraints.reference. citedResponsibleParty	/mdb:MD_Metadata/mdb:identificationInfo/*/mri:resourceConstraints/ mco:MD_LegalConstraints/mco:reference/cit:CI_Citation/ cit:citedResponsibleParty/cit:CI_Responsibility	
copyright Year	Number	The year during which the claimed copyright for the CreativeWork was first asserted.	identificationInfo.resourceConstraints.reference. date[dateType =’publication’].date	/mdb:MD_Metadata/mdb:identificationInfo/*/mri:resourceConstraints/ mco:MD_LegalConstraints/mco:reference/cit:CI_Citation/ cit:date/cit:CI_Date[cit:dateType/cit:CI_DateTypeCode =’publication’]/cit:dateType	
creator	Organization or Person	The creator/author of this CreativeWork. This is the same as the Author property for CreativeWork.	identificationInfo.citation.citedResponsibleParty [role =’author’].party.name or identificationInfo.citation.citedResponsibleParty[role =’originator’].party.namea	/mdb:MD_Metadata/mdb:identificationInfo/*/mri:citation/cit:CI_Citation/ cit:citedResponsibleParty/cit:CI_Responsibility[cit:role/cit:CI_RoleCode =’author’] or /mdb:MD_Metadata/mdb:identificationInfo/*/mri:citation/cit:CI_Citation/ cit:citedResponsibleParty/cit:CI_Responsibility[cit:role/cit:CI_RoleCode =’originator’]	
date Created	Date or DateTime	The date on which the CreativeWork was created or the item was added to a DataFeed.	identificationInfo.citation.date[dateType =’creation’]. datea	/mdb:MD_Metadata/mdb:identificationInfo/*/mri:citation/ cit:CI_Citation/cit:date/cit:CI_Date[cit:dateType/cit:CI_DateTypeCode =’creation’]/cit:date/gco:DateTime	
date Modified	Date or DateTime	The date on which the CreativeWork was most recently modified or when the item’s entry was modified within a DataFeed.	identificationInfo.citation.date[dateType =’revision’]. datea	/mdb:MD_Metadata/mdb:identificationInfo/*/mri:citation/cit:CI_Citation/ cit:date/cit:CI_Date[cit:dateType/cit:CI_DateTypeCode =’revision’]/cit:date/gco:DateTime	
date Published	Date	Date of first broadcast/publication.	identificationInfo.citation.date[dateType =’publication’].datea	/mdb:MD_Metadata/mdb:identificationInfo/*/mri:citation/cit:CI_Citation/ cit:date/cit:CI_Date[cit:dateType/cit:CI_DateTypeCode =’publication’]/cit:date/gco:Date	
editor	Person	Specifies the person who edited the CreativeWork.	identificationInfo.citation.citedResponsibleParty [role =’editor’].party.namea	/mdb:MD_Metadata/mdb:identificationInfo/*/mri:citation/cit:CI_Citation/ cit:citedResponsibleParty/ cit:CI_Responsibility[cit:role/cit:CI_RoleCode =’editor’]	
encoding	MediaObject	A media object that encodes this CreativeWork. This property is a synonym for associatedMedia. Supersedes encodings.			
fileFormat	Text or URL	Media type, typically MIME format (see IANA site) of the content e.g., application/zip of a SoftwareApplication binary. In cases where a CreativeWork has several media type representations, ’encoding’ can be used to indicate each MediaObject alongside particular fileFormat information. Unregistered or niche file formats can be indicated instead via the most appropriate URL, e.g., defining Web page or a Wikipedia entry.	identificationInfo.resourceFormat. formatSpecificationCitation	/mdb:MD_Metadata/mdb:identificationInfo/mri:MD_DataIdentification/ mri:resourceFormat/mrd:MD_Format/mrd:formatSpecificationCitation/ cit:CI_Citation	
funder	Organization or Person	A person or organization that supports (sponsors) something through some kind of financial contribution.	identificationInfo.citation.citedResponsibleParty [role =’funder’].party.namea	/mdb:MD_Metadata/mdb:identificationInfo/*/mri:citation/cit:CI_Citation/ cit:citedResponsibleParty/cit:CI_Responsibility[cit:role/cit:CI_RoleCode =’funder’]	
hasPart	CreativeWork	Indicates a CreativeWork that is (in some sense) a part of this CreativeWork. Reverse property isPartOf	identificationInfo.associatedResource [associationType =’isComposedOf’].namea	/mdb:MD_Metadata/mdb:identificationInfo/*/mri:associatedResource/ mri:MD_AssociatedResource[mri:associationType/mri:DS_AssociationTypeCode =’isComposedOf’]/mri:name/cit:CI_Citation	
isAccessibleForFree	Boolean	A flag to signal that the publication is accessible for free.	distributionInfo.distributionFormat.format Distributor.distributionOrderProcess.fees	/mdb:MD_Metadata/mdb:distributionInfo/mrd:MD_Distribution/ mrd:distributionFormat/mrd:MD_Format/mrd:formatDistributor/ mrd:MD_Distributor/mrd:distributionOrderProcess/ mrd:MD_StandardOrderProcess/mrd:fees	
isPartOf	CreativeWork	Indicates a CreativeWork that this CreativeWork is (in some sense) part of. Reverse property hasPart	identificationInfo.associatedResource [associationType =’LargerWorkCitation’].namea	/mdb:MD_Metadata/mdb:identificationInfo/*/mri:associatedResource/ mri:MD_AssociatedResource[mri:associationType/mri:DS_AssociationTypeCode =’LargerWorkCitation’]/mri:name/cit:CI_Citation	
keywords	Text	Keywords or tags used to describe this content. Multiple entries in a keywords list are typically delimited by commas.	identificationInfo.descriptiveKeywords[type =’theme’]. Keyworda	/mdb:MD_Metadata/mdb:identificationInfo/*/mri:descriptiveKeywords/ mri:MD_Keywords[mri:type/mri:MD_KeywordTypeCode =’theme’]/mri:keyword/gco:CharacterString	
license	CreativeWork or URL	A license document that applies to this content, typically indicated by URL.	identificationInfo.resourceConstraints.reference	/mdb:MD_Metadata/mdb:identificationInfo/mri:MD_DataIdentification/ mri:resourceConstraints/mco:MD_LegalConstraints/mco:reference/cit:CI_Citation	
position	Integer or Text	The position of an item in a series or sequence of items. (While schema.org considers this a property of CreativeWork, it is also the way to indicate ordering in any list (e.g., the Authors list). By default arrays are unordered in JSON-LD			
producer	Organization or Person	The person or organization who produced the work (e.g., music album, movie, tv/radio series etc.).	identificationInfo.citation.citedResponsibleParty [role =’creator’].party.namea	/mdb:MD_Metadata/mdb:identificationInfo/*/mri:citation/cit:CI_Citation/ cit:citedResponsibleParty/cit:CI_Responsibility[cit:role/cit:CI_RoleCode =’creator’]/cit:party/a	
provider	Organization or Person	The service provider, service operator, or service performer; the goods producer. Another party (a seller) may offer those services or goods on behalf of the provider. A provider may also serve as the seller. Supersedes carrier.	identificationInfo.pointOfContact [role =’pointOfContact’].party.namea	/mdb:MD_Metadata/mdb:identificationInfo/srv:SV_ServiceIdentification/ mri:pointOfContact/cit:CI_Responsibility[cit:role/cit:CI_RoleCode =’provider’]/cit:party/*	
publisher	Organization or Person	The publisher of the creative work.	identificationInfo.citation.citedResponsibleParty [role =’publisher’].party.namea	/mdb:MD_Metadata/mdb:identificationInfo/*/mri:citation/cit:CI_Citation/ cit:citedResponsibleParty/cit:CI_Responsibility[cit:role/cit:CI_RoleCode =’publisher’]/cit:party/a	
sponsor	Organization or person	A person or organization that supports a thing through a pledge, promise, or financial contribution. e.g., a sponsor of a Medical Study or a corporate sponsor of an event.	identificationInfo.citation.citedResponsibleParty [role =’sponsor’].party.namea	/mdb:MD_Metadata/mdb:identificationInfo/*/mri:citation/cit:CI_Citation/ cit:citedResponsibleParty/cit:CI_Responsibility[cit:role/cit:CI_RoleCode =’sponsor’]/cit:party/a	
version	Number or Text	The version of the CreativeWork embodied by a specified resource.	identificationInfo.citation.edition	/mdb:MD_Metadata/mdb:identificationInfo/mri:MD_DataIdentification/mri:citation/ cit:CI_Citation/cit:edition/gco:CharacterString	
Notes.

a Multiple CodeMeta terms are mapped to this ISO XML element, some with different attributes.

Schema:Thing.CreativeWork.SoftwareSourceCode

The Thing.CreativeWork.SoftwareSourceCode schema provides a vocabulary for describing computer programming source code. This mapping includes four items listed in Table 7.

Table 7 Mapping of CodeMeta terms from the Thing.CreativeWork.SoftwareSourceCode schema to ISO 19115-1 and ISO 19115-3.

Property	Type	Description	ISO 19115-1	ISO 19115-3	
Code Repository	URL	Link to the repository where the un-compiled, human readable code and related code is located (SVN, github, CodePlex).	distributionInfo.distributor.distributor TransferOptions.onLine[function =’download’].linkage or distributionInfo.distributor.distributorTransferOptions. onLine[function =’information’].linkage or distributionInfo.transferOptions.onLine [function =’download’].linkage or distributionInfo.transferOptions.onLine [function =’information’].linkage or identificationInfo.citation.onlineResource [function =’download’].linkage or identificationInfo.citation.onlineResource [function =’information’].linkage*	/mdb:MD_Metadata/mdb:distributionInfo/mrd:MD_Distribution/ mrd:distributor/mrd:MDDistributor/mrd:distributorTransferOptions/ mrd:MDDigitalTransferOptions/mrd:onLine/cit:CI_OnLineResource[cit: function/cit:CI_OnLineFunctionCode =’information’]/ cit:linkage/gco:CharacterString or /mdb:MD_Metadata/ mdb:distributionInfo/mrd:MD_Distribution/mrd:transferOptions/ mrd:MDDigitalTransferOptions/mrd:onLine/cit:CI_OnLineResource [cit:function/cit:CI_OnLineFunctionCode =’information’]/ cit:linkage/gco:CharacterString or /mdb:MD_Metadata/ mdb:identificationInfo/*/mri:citation/cit:CI_Citation/ cit:onlineResource/cit:CI_OnLineResource[cit:function/ cit:CI_OnLineFunctionCode =’information’] /cit:linkage/ gco:CharacterString or /mdb:MD_Metadata/mdb:distributionInfo/ mrd:MD_Distribution/mrd:distributor/mrd:MDDistributor/ mrd:distributorTransferOptions/mrd:MDDigitalTransferOptions/ mrd:onLine/cit:CI_OnLineResource[cit:function/ cit:CI_OnLineFunctionCode =’download’] /cit:linkage/ gco:CharacterString or /mdb:MD_Metadata/mdb:distributionInfo/ mrd:MD_Distribution/mrd:transferOptions/mrd:MDDigitalTransferOptions/ mrd:onLine/cit:CI_OnLineResource[cit:function/ cit:CI_OnLineFunctionCode =’download’] /cit:linkage/ gco:CharacterString or /mdb:MD_Metadata/mdb:identificationInfo/*/ mri:citation/cit:CI_Citation/cit:onlineResource/ cit:CI_OnLineResource[cit:function/ cit:CI_OnLineFunctionCode =’download’] /cit:linkage/gco:CharacterString	
ProgrammingLanguage	Computer Language or Text	The computer programming language.	identificationInfo.descriptiveKeywords[type =’theme’]. keyworda	/mdb:MD_Metadata/mdb:identificationInfo/*/mri:descriptiveKeywords/ mri:MD_Keywords[mri:type/mri:MD_KeywordTypeCode =’theme’]/mri:keyword/gco:CharacterString	
Runtime Platform	Text	Runtime platform or script interpreter dependencies (Example—Java v1, Python2.3, .Net Framework 3.0). Supersedes runtime.	identificationInfo.environmentDescriptiona	/mdb:MD_Metadata/mdb:identificationInfo/mri:MD_DataIdentification/ mri:environmentDescription/gco:CharacterString	
Target Product	Software Application	Target Operating System/Product to which the code applies. If applies to several versions, just the product name can be used.	identificationInfo.associatedResource.namea	/mdb:MD_Metadata/mdb:identificationInfo/*/mri:associatedResource/ mri:MD_AssociatedResource/mri:name/cit:CI_Citation	
Notes.

a Multiple CodeMeta terms are mapped to this ISO XML element, some with different attributes.

CodeMeta:SoftwareSourceCode

The CodeMeta:SoftwareSourceCode schema extends Thing.CreativeWork.SoftwareSourceCode with terms created by the CodeMeta Project. This mapping includes ten items listed in Table 8.

Table 8 Mapping of CodeMeta terms from the CodeMeta.SoftwareSourceCode schema to ISO 19115-1 and ISO 19115-3.

Property	Type	Description	ISO 19115-1	ISO 19115-3	
Build instructions	URL	Link to installation instructions/documentation	IdentificationInfo.additionalDocumentationa	/mdb:MD_Metadata/mdb:identificationInfo/*/ mri:additionalDocumentation/cit:CI_Citation	
cont Integration	URL	link to continuous integration service	identificationInfo.additionalDocumentationa	/mdb:MD_Metadata/mdb:identificationInfo/*/ mri:additionalDocumentation/cit:CI_Citation	
developmentStatus	Text	Description of development status, e.g., Active, inactive, supsended. See repostatus.org	identificationInfo.status	/mdb:MD_Metadata/mdb:identificationInfo/*/ mri:status/mcc:MD_ProgressCode	
embargo Date	Date	Software may be embargoed from public access until a specified date (e.g., pending publication, 1 year from publication)	identificationInfo.citation.date[dateType =’released’]. datea	/mdb:MD_Metadata/mdb:identificationInfo/*/mri:citation/ cit:CI_Citation/cit:date/cit:CI_Date[cit:dateType/ cit:CI_DateTypeCode =’released’]/cit:date/gco:DateTime	
funding	Text	Funding source (e.g., specific grant)	identificationInfo.associatedResourcea	/mdb:MD_Metadata/mdb:identificationInfo/*/ mri: associatedResource/mri:MD_AssociatedResource/mri:name/cit:CI_Citation	
issueTracker	URL	link to software bug reporting or issue tracking system	identificationInfo.resourceSpecificUsage. identifiedIssues.onlineResource.linkage	/mdb:MD_Metadata/mdb:identificationInfo/*/mri: resourceSpecificUsage/mri:MD_Usage/mri:identifiedIssues/cit:CI_Citation	
maintainer	Person	Individual responsible for maintaining the software (usually includes an email contact address)	identificationInfo.pointOfContact	/mdb:MD_Metadata/mdb:identificationInfo/*/mri:pointOfContact	
readme	URL	link to software Readme file	identificationInfo.additionalDocumentationa	/mdb:MD_Metadata/mdb:identificationInfo/*/ mri:additionalDocumentation/cit:CI_Citation	
reference Publication	ScholarlyArticle	An academic publication related to the software.	identificationInfo.additionalDocumentationa	/mdb:MD_Metadata/mdb:identificationInfo/*/ mri:additionalDocumentation/cit:CI_Citation	
software Suggestions	SoftwareSourceCode	Optional dependencies, e.g., for optional features, code development, etc	identificationInfo.additionalDocumentationa	/mdb:MD_Metadata/mdb:identificationInfo/*/ mri:additionalDocumentation/cit:CI_Citation	
Notes.

a Multiple CodeMeta terms are mapped to this ISO XML element, some with different attributes.

Schema:Thing.CreativeWork.SoftwareApplication

The Thing.CreativeWork.SoftwareApplication schema provides a vocabulary for describing a software application. This mapping includes fifteen items listed in Table 9.

Table 9 Mapping of CodeMeta terms from the schema:Thing.CreativeWork.SoftwareApplication schema to ISO 19115-1 and 19115-3.

Property	Type	Description	ISO 19115-1	ISO 19115-3	
application Category	Text or URL	Type of software application, e.g., ’Game, Multimedia’.	identificationInfo.descriptiveKeywords[type =’theme’].keyword*	/mdb:MD_Metadata/mdb:identificationInfo/*/mri:descriptiveKeywords/ mri:MD_Keywords[mri:type/mri:MD_KeywordTypeCode =’theme’]/ mri:keyword/gco:CharacterString	
application SubCategory	Text or URL	Subcategory of the application, e.g., ‘Arcade Game’.	identificationInfo.descriptiveKeywords[type =’theme’].keyword*	/mdb:MD_Metadata/mdb:identificationInfo/*/mri:descriptiveKeywords/ mri:MD_Keywords[mri:type/mri:MD_KeywordTypeCode =’theme’]/mri:keyword/ gco:CharacterString	
download Url	URL	If the file can be downloaded, URL to download the binary.	distributionInfo.distributor.distributor TransferOptions.onLine[function =’download’].linkage or distributionInfo.transfer Options.onLine[function =’download’].linkage or identificationInfo.citation.onlineResource[function =’download’].linkage	/mdb:MD_Metadata/mdb:distributionInfo/mrd:MD_Distribution/ mrd:distributor/mrd:MD_Distributor/mrd:distributorTransferOptions/ mrd:MD_DigitalTransferOptions/mrd:onLine/cit:CI_OnlineResource [cit:function/cit:CI_OnLineFunctionCode =’download’]/ cit:linkage/gco:CharacterString or /mdb:MD_Metadata/ mdb:distributionInfo/mrd:MD_Distribution/mrd:transferOptions/ mrd:MD_DigitalTransferOptions/mrd:onLine/cit:CI_OnlineResource [cit:function/cit:CI_OnLineFunctionCode =’download’] / cit:linkage/gco:CharacterString or /mdb:MD_Metadata/ mdb:identificationInfo/*/mri:citation/cit:CI_Citation/ cit:onlineResource/cit:CI_OnlineResource[cit:function/ cit:CI_OnLineFunctionCode =’download’] /cit:linkage/gco:CharacterString	
fileSize	Text	Size of the application/package (e.g., 18MB). In the absence of a unit (MB, KB etc.), KB will be assumed.	distributionInfo.transferOptions.transferSize or distributionInfo.distributionFormat.formatDistributor. distributorTransferOptions.transferSize or distributionInfo.distributor.distributorTransferOptions .transferSize	/mdb:MD_Metadata/mdb:distributionInfo/mrd:MD_Distribution/ mrd:transferOptions/mrd:MD_DigitalTransferOptions/mrd:transferSize/ gco:Real or /mdb:MD_Metadata/mdb:distributionInfo/mrd:MD_Distribution/ mrd:distributionFormat/mrd:MD_Format/mrd:formatDistributor/ mrd:MD_Distributor/mrd:distributorTransferOptions/ mrd:MD_DigitalTransferOptions/mrd:transferSize/gco:Real or / mdb:MD_Metadata/mdb:distributionInfo/mrd:MD_Distribution/ mrd:distributor/mrd:MD_Distributor/mrd:distributorTransferOptions/ mrd:MD_DigitalTransferOptions/mrd:transferSize/gco:Real	
installUrl	URL	URL at which the app may be installed, if different from the URL of the item.	distributionInfo.distributor.distributor TransferOptions.onLine[function =’download’]. linkage or distributionInfo.transferOptions.onLine [function =’download’].linkage or identificationInfo. citation.onlineResource[function =’download’].linkage	/mdb:MD_Metadata/mdb:distributionInfo/mrd:MD_Distribution/ mrd:distributor/mrd:MD_Distributor/mrd:distributorTransferOptions/ mrd:MD_DigitalTransferOptions/mrd:onLine/cit:CI_OnlineResource [cit:function/cit:CI_OnLineFunctionCode =’download’]/ cit:linkage/gco:CharacterString or /mdb:MD_Metadata/ mdb:distributionInfo/mrd:MD_Distribution/mrd:transferOptions/ mrd:MD_DigitalTransferOptions/mrd:onLine/cit:CI_OnlineResource [cit:function/cit:CI_OnLineFunctionCode =’download’]/ cit:linkage/gco:CharacterString or /mdb:MD_Metadata/ mdb:identificationInfo/*/mri:citation/cit:CI_Citation/ cit:onlineResource/cit:CI_OnlineResource[cit:function/ cit:CI_OnLineFunctionCode =’download’]/cit:linkage/gco:CharacterString	
memory Requirements	Text or URL	Minimum memory requirements.	identificationInfo.environmentDescriptiona	/mdb:MD_Metadata/mdb:identificationInfo/mri: MD_DataIdentification/mri:environmentDescription/gco:CharacterString	
operating System	Text	Operating systems supported (Windows 7, OSX 10.6, Android 1.6).	identificationInfo.environmentDescriptiona	/mdb:MD_Metadata/mdb:identificationInfo/mri: MD_DataIdentification/mri:environmentDescription/gco:CharacterString	
permissions	Text	Permission(s) required to run the app (for example, a mobile app may require full internet access or may run only on wifi).	identificationInfo.resourceConstraints or identificationInfo.resourceConstraints.reference. onlineResource.linkage	/mdb:MD_Metadata/mdb:identificationInfo/*/mri:resourceConstraints/ mco:MD_LegalConstraints or /mdb:MD_Metadata/mdb:identificationInfo/ mri:MD_DataIdentification/mri:resourceConstraints/ mco:MD_LegalConstraints/mco:reference/cit:CI_Citation/ cit:onlineResource/cit:CI_OnlineResource/cit:linkage	
processor Requirements	Text	Processor architecture required to run the application (e.g., IA64).	identificationInfo.environmentDescriptiona	/mdb:MD_Metadata/mdb:identificationInfo/mri: MD_DataIdentification/mri:environmentDescription/gco:CharacterString	
release Notes	Text or URL	Description of what changed in this version.	identificationInfo.additionalDocumentationa	/mdb:MD_Metadata/mdb:identificationInfo/*/mri:additionalDocumentation/ cit:CI_Citation	
software Help	CreativeWork	Software application help.	identificationInfo.additionalDocumentationa	/mdb:MD_Metadata/mdb:identificationInfo/*/mri:additionalDocumentation/ cit:CI_Citation	
software Requirements	SoftwareSourceCode	Required software dependencies	identificationInfo.additionalDocumentationa	/mdb:MD_Metadata/mdb:identificationInfo/*/mri:additionalDocumentation/ cit:CI_Citation	
software Version	Text	Version of the software instance.	identificationInfo.citation.edition	/mdb:MD_Metadata/mdb:identificationInfo/mri:MD_DataIdentification/ mri:citation/cit:CI_Citation/cit:edition	
storage Requirements	Text or URL	Storage requirements (free space required).	identificationInfo.environmentDescriptiona	/mdb:MD_Metadata/mdb:identificationInfo/mri:MD_DataIdentification/ mri:environmentDescription/gco:CharacterString	
supporting Data	DataFeed	Supporting data for a Software Application.	identificationInfo.associatedResource.namea	/mdb:MD_Metadata/mdb:identificationInfo/*/mri:associatedResource/ mri:MD_AssociatedResource/mri:name/cit:CI_Citation	
Notes.

a Multiple CodeMeta terms are mapped to this ISO XML element, some with different attributes.

Conclusions

The ISO metadata standards were originally developed by ISO Technical Committee 211 to serve as the standard and structured part of the documentation needed to discover, access, use, and understand datasets. The standards acknowledge that they are generic, and they include several mechanisms for extension to address specific needs of communities that build on the standards. The generic nature of these standards is reflected in the breadth of the codelist that can be used to describe the scope of a particular metadata record (see list in Model Characteristics Section above and Habermann, 2018a; Habermann, 2018b; Habermann, 2018c).

The CodeMeta project recently proposed over sixty terms that can be used in metadata for software. This recommendation provides a framework that provides insight into what ISO metadata for software might contain. The ISO metadata standards include several elements that directly cite software, e.g., the processingInformation.softwareReference or algorithm.citation elements, and the primary purpose of the mappings proposed here is to support ISO users that want to (1) express software metadata in a dialect that they are familiar with and (2) to facilitate translation of software metadata written using ISO standards to CodeMeta-compliant JSON-LD.

Note that the purpose is different than the primary purpose of the crosswalks proposed on the CodeMeta project site which is to facilitate automated translations between different JSON-LD vocabularies and RDF. Adding an ISO crosswalk to the CodeMeta framework was the original goal of this work, but the differences identified and described here are significant enough to make that impossible. This translation may be facilitated with an XSLT built specifically for that purpose, like the one created by Peroni, Lapeyre & Shotton, 2012, but that is beyond the scope of this initial comparison.

The process of creating a mapping between these two representations surfaced some differences that complicate the mapping. Some of these differences are related to hard and soft typing used in the two models and others are related to increased flexibility that is required in a generic standard like ISO for documenting citations, distribution channels, and related resources.

The differences in approach described here probably apply to many mappings from XML to RDF representations. Peroni, Lapeyre & Shotton, 2012 identified and discussed some similar challenges when mapping between JATS XML and SPAR Ontologies. They attributed these differences to “differing philosophical viewpoints for XML and RDF”, then described two reasons that XML element names are ambiguous when used in isolation: attributes and hierarchical structure. They suggest that the JATS standard (and by implication all XML representations) is “deliberately vague” and that the hierarchical structure of XML is “not formalized and implicitly lives outside the XML schema of the language”. It is certainly true that XML element names can be ambiguous without associated attributes and hierarchical structure but ignoring those methods of providing meaning in XML and then stating that elements alone are ambiguous does not contribute to understanding differences between how these two approaches describe resources.

I examine these differences in more detail and offer an explanation that is related to different approaches to defining types: XML can use attributes and structure to provide information about types and meaning while RDF must rely on unambiguous and shared definitions of terms. To use the case described in Peroni, Lapeyre & Shotton, 2012, the JATS element “article” is definitely ambiguous. That is why the definition of the element includes an attribute called article-type to clarify the type of the article being described. Details of the importance of attributes for semantics in XML is described in detail by Seligy (2018) along with potential problems that result from ignoring attribute values.

Peroni, Lapeyre & Shotton (2012) also acknowledge the requirement for unambiguous definitions that are shared across communities. They state that “A cornerstone of the Semantic Web is the use of open published ontologies to give precise and universally available definitions to terms, so that RDF statements, whatever else they are, are unambiguous in their meaning.” While this may be a goal of ontology development efforts, many of the definitions currently used in CodeMeta and schema.org (shown in Tables 5–9) are unclear, non-unique, and ambiguous.

ISO mappings are proposed for sixty-four of the sixty-eight CodeMeta V2 terms. These mappings can be used to create CodeMeta-compliant metadata from existing stores of ISO metadata and to add CodeMeta compliant software citations in the future. This compares to an average of 11.2 mappings for other dialects included in the CodeMeta crosswalk. This disparity reflects the use cases targeted by the dialects. Many of the dialects that have been mapped to CodeMeta are focused on citation or dependency identification and management while ISO and CodeMeta share additional targets that include access, use, and understanding.

Supplemental Information

Supplemental Information 1 Mapping Tables 4–9 as machine readable csv

The mappings include the schemas, property names, types, and descriptions from the CodeMeta vocabulary, conceptual paths for the ISO items (ISO 19115-1), and XPaths from the standard XML representation (ISO 19115-3).

Click here for additional data file.

Thanks to Matt Jones, Carl Boettiger, Dennis Walworth, and Melissa Harrison for helpful comments and discussion of the initial draft of this paper.

Additional Information and Declarations

Competing Interests

Author Contributions

Data Availability

Ted Habermann is the Director of Earth Science at The HDF Group. The author declares that there are no competing interests.

Ted Habermann conceived and designed the experiments, performed the experiments, analyzed the data, prepared figures and/or tables, performed the computation work, authored or reviewed drafts of the paper, approved the final draft.

The following information was supplied regarding data availability:

Habermann, Ted (2018): MappingTables_ISO19115-1ToCodeMeta.csv. figshare. Dataset. https://doi.org/10.6084/m9.figshare.7430282.v1.

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
