# Peer review of "Mapping ISO 19115-1 geographic metadata standards to CodeMeta"

_PeerJ Computer Science, doi:10.7717/peerj-cs.174_

## Round 0.1 · original submission · Major Revisions

I have carefully evaluated the reviews of the article and, since quite contradictory suggestions have been returned by the reviewers, I've read the article as well so as to have a clearer position on the work done and the possible issues.

While I'm not in favour of rejecting it, I do think an appropriate (I would say huge) extension of the text and the content of the article should be provided before considering this work accepted for publication. Both the reviewers (even the favourable one) have highlighted important issues that must be addressed. I would suggest the author keep all of them in careful consideration for the revise his article. In particular, I see at least three major issues:

1. There is a quite extensive literature available that describes the mapping between different specifications, such as (disclosure: I'm one of the authors of some of the following):

- Peroni, S., Lapeyre, D. A., Shotton, D. (2012). Mapping JATS to RDF using the SPAR (Semantic Publishing and Referencing) Ontologies. In Proceeding of the Journal Article Tag Suite Conference 2012 (JATS-Con 2012). Bethesda, Maryland, USA: National Center for Biotechnology Information. http://www.ncbi.nlm.nih.gov/books/NBK100491/

- the efforts at Metadata2020 (http://www.metadata2020.org)

It would be great to have a discussion of similar kinds of works and the main issues that they had to address, in order to understand if some of those are general to be found also in any tentative XML to RDF alignment specification - like the one proposed by the author.

2. The distinction between the so-called "hard-typing" and "soft-typing" is very interesting, but should be clarified better (providing examples), in particular, if it is a typical situation that happens in XML-to-RDF alignments - and I believe it does. In addition, I agree with the second reviewer that that "hard" and "soft" nomenclature is not actually shared and accepted.

3. The article would need to include more discussion about the proposed mapping from at least two perspectives: (a) for each table, it would be good to have a clear description of the main alignment issues, so as to understand clearly which choices have been done to address them and why; and (b) a validation (as proposed by the second reviewer) is necessary to publish such alignment tables in a journal publication.

Other minor points identified, in addition to those ones introduced by the reviewers:

- In the Introduction, "(ISO 19115-1)" should be a proper citation, not just a link.

- I could not find any citation to Codemeta in the introduction.

- The structure and content of the table at "https://github.com/codemeta/codemeta/blob/master/crosswalk.csv" should be clearly explained in the article. Is it a contribution of the article, or something that has been made available as additional material?

- "The most commonly used representation of the ISO standards is XML" -> Examples should be added when explicit XML documents/schemas and related XPath queries are mentioned and used.

- "The ISO model is soft-typed and the Codemeta model is hard-typed. Differences related to these 137 different approaches are listed in Table 2 and described below." -> I cannot see a clear explanation of the hard vs soft issue in the table, honestly.

- The definition of the three kinds of citations is not clear. Examples are needed in order to better understand the differences between the three. In addition, examples of representation are necessary, for both ISO and Codemeta Citations.

I would like to see these kinds of articles published in PeerJ CS, but honestly, this contribution needs further improvement to deserve to be accepted in the journal. Thus, I would suggest having a "major revision" as the final decision. It is worth mentioning that it would be possible that I will decide to involve an additional reviewer for checking the revision that will be provided.

Thanks again for submitting your contribution to PeerJ CS.
Have a nice day :-)

S.

Reviewer 1 ·

Basic reporting

The manuscript is well written, appropriately referenced, and clearly structured. The background supplied is sufficient if somewhat minimal. This is largely a stylistic choice which leaves the manuscript focused on the details of the research performed, but it could nonetheless be valuable to set these issues more deliberately into their broader context. The issues discussed here are not unique to the relatively niche "codemeta" description and the ISO standard, but appear to fall into broader distinctions between the underlying, widely used Schema.org standard and many ISO standards.

Experimental design

no comment.

Validity of the findings

The mapping presented here is impressively comprehensive and of obvious immediate value to the relevant communities in this space. These results are clearly presented, though a machine-readable (e.g. csv file) version of the tables presented in the pdf would no doubt be welcome.

I think the one area I will comment on is the discussion of the distinction between ISO and "codemeta" formats. I don't think this impacts the validity of the author's conclusion, but perhaps it merits some discussion. In any event, I include it here in case it is helpful and not merely confusing.


While I agree with the author's insightful observation that a difference between what the author calls "hard types" vs "soft types" is precisely what distinguishes between these, it seems to me that this distinction reflects more fundamental differences in the data model involved. Schema.org explicitly follows the Resource Description Framework (RDF) of subject-predicate-object triples, often expressed in JSON-LD as in codemeta. The "hard typing" arises because "author" is the natural predicate in the triple such as: This "software" has "author" "Tim Berners-Lee", e.g.

{
"@type": "SoftwareSourceCode",
"author": "Tim Berners-Lee"
}


The equivalent statement in the ISO standard is typically expressed in using an XML structure that does not permit the same simple semantics, in particular, because XML has a notion of attributes that can be used to modify the meaning of a node (e.g. citedResponsibleParty[role='author'] which is not available in JSON and is correspondingly more ambiguous to express in RDF triples. While we like to think of both standards as abstractions independent of serialization, it is hard not to notice the role of those serializations here. I am not an expert in these matters, and I suspect it that the ISO standard could be expressed as RDF, but it does appear to me that the difference between 'hard' and 'soft' are really differences between a vocabulary intended as Linked Data and a standard intended to be specified as an XML schema.

Reviewer 2 ·

Basic reporting

The article proposes a mapping between ISO metadata (from technical committee
211) and CodeMeta metadata. The writing is clear and professional. References
are very scarce, and should definitely be improved. The article is
self-contained and easy to follow.

Experimental design

While, strictly speaking, no "experimental design" is applicable to this
contribution, the main problem of the paper is precisely in its validation.
The proposed mapping seems to be, in essence, the author's opinion of how the
two ontologies should be related (see below to some potential issues with
author's choices, but the point here is more general). Given how CodeMeta
works, and in particular how their crosswalk table is maintained, it seems to
this reviewer that the appropriate course of action to validate author's
choices is to propose these mappings to the CodeMeta community, to gather
feedback (and potentially criticism) on the proposed mapping. A scientific
publication does not seem warranted without that essential validation step.

Validity of the findings

- line 58: I don't understand the dichotomy in requirements between
discovering/citing software and use/trust software, The classification of
metadata requirements can be much richer than that and this partition should
be explained

- line 131: Hard and soft typing isn't a reference to existing terminology,
would be interesting to know where it comes from. The governance issue for
soft types seems questionable, because even attribute types (in soft typing)
can be subject to validation using various XML schema languages

- line 161: print issue where the items aren't found

- line 254: the author should reference the hash of the file and not the
unstable url provided

- table 1 should be versioned/dated because schema.org can make changes and
doesn't use versions

Regarding the proposed mappings:

- relatedLink and sameAS properties are mapped to the same ISO property

- contributor is mapped to an ISO property where it excludes the role=author
from the list of citedResponsibleParty instead of using role=contributor (In
my understanding from the paper of soft typing, this addition of
role=contributor shouldn't be a problem)

- producer property is mapped to citedResponsibleParty role=creator - why not
role=producer?

- creator property is mapped to citedResponsibleParty role=originator- why not
role=creator?

- codeRepository property, which is one of the most source code specific
properties, is mapped to 5 different options, but none is specifically the
characteristics of a code repository

- programmingLanguage is mapped to descriptiveKeyword[type='theme'], but there
is a keywords property in CodeMeta- this double mapping and other properties
that are mapped with the same value should be explained or commented

- softwareSuggestions is mapped to additionalDocumentation but
softwareSuggestions is refering to optional dependencies and software
components and not documentation.

- 8 CodeMeta properties are mapped to additionalDocumentation - this should be
explained or commented.

The mapping is from CodeMeta to ISO and should be specified as a 1 to n
properties, if an automated process took a codemeta.json file and converted it
to ISO, this mapping would be the best way not to loose information, on the
other hand if the process was reversed the data would be duplicated and quite
frankly hard to comprehend. This mapping results in a very high "dialect
coverage" because of the 1 to n links from CodeMeta to ISO, it would be
interesting to know what is the "dialect coverage" if the other way was mapped
from ISO to CodeMeta.

Also in the mapping the description of the property there is only the CodeMeta
description and it might be helpful to have the description of the ISO
properties. If the paper's objective is to open the discussion with the
CodeMeta community before adding the ISO vocabulary to the crosswalk, I would
suggest to state it clearly in the abstract. Otherwise this mapping should be
added with a PR to the crosswalk table before publication.

Final remark, examples of mapping difficulties and resolution would be a great
way to explain some mapping choices.

Additional comments

Nothing else to report.

---

## Round 0.2 · Minor Revisions

Dear author,

As I anticipated in the previous decision letter, I decided to involve an additional reviewer for checking the new revision provided. This new reviewer had access to all the material available since the first submission, including the reviews already provided by the other reviewers, and she considered all this existing inherited knowledge in her review as well. In addition, she provided also a fresh reading to your content, so as to spot some additional minor concerns that should be addressed appropriately.

After reading the new revision and the new review, I'm very confident that this article should be accepted to PeerJ CS. However, there are still minor issues that should be addressed appropriately before publishing it. In addition to the issues indicated by the new reviewer, there are additional points that I would like to highlight:

1. You said that you have assessed and, thus, validated the mapping by involving directly the Codemeta Project Leaders, which is good. However, while having them mentioned in the acknowledgments is fine, I would also like to see a few additional paragraphs in the paper where you explain precisely the whole process you followed for creating and then validating the alignment you propose. This is useful since it would provide a methodology that can be followed in the future for repeating the alignment with new versions of the standards [= reproducibility], and that can even be adopted in similar mapping tasks among different standards [= reusability].

2. You suggested that "the challenges of the mapping were interesting enough to get them out into the discussion before taking on a transform", and thus claim that the development of an XSLT mapping document goes beyond the current scope of this article. However – according to my personal experience in these kinds of mapping tasks (see Peroni et al., 2012) – the development of such mapping rules by means of a particular transformation template, such as an XSLT document, was very helpful so as to catch specific issues that were not identified in the mapping -- see, for instance, the section "Using XSLT to automate the conversion from JATS XML to RDF" in (Peroni et al., 2012). In addition, I strongly believe that to have such XSLT (or an any other executable transformation document, with examples) would be an important and, honestly, mandatory contribution for this article. Having such file would, in principle, also address the reviewer's comment about the availability of the mapping in machine-readable formats.

3. Some typos:
- "xPath" -> "XPath"
- "SPAR ontologies" -> "SPAR Ontologies"

Please consider to address all the aforementioned points and all the ones raised by the new reviewer before submitting a new revision of the article.

Thanks again for submitting your contribution to PeerJ CS – I'm really looking forward to having your revised manuscript.
Have a nice day :-)

S.

Reviewer 3 ·

Basic reporting

The paper "Mapping ISO metadata standards to codemeta" describes the challenges of mapping the ISO 19115-1
metadata standard to the codemeta project, which defines a vocabulary for software metadata.

As indicated by previous reviews, the paper is well-written and clear. In terms of the background provided, it seems
sufficient for a reader with some knowledge in this area. A bit more introduction about terminology and formats
might be needed for readers not familiar with metadata concepts in some places (e.g. explain the acronym RDF and RDF model)
to make the manuscript self-contained.

This new version of the paper has addressed most comments by reviewers and the paper is improved.

Here a few more comments and suggestions for your consideration.

The title of the paper refers to the "ISO metadata", but I recommend to change it to refer explicitly to the ISO 19115-1 standard
for geographic information (https://www.iso.org/standard/53798.html), making clear that it was designed for geospatial data metadata.
While many of the elements of the standards are generic (as also emphasized in the conclusions), I think it is important to make a
better distinction between the purpose of the creation of ISO 19115 vs the purpose for creating codemeta, in particular. This should
also help in describing the domains in which both standards are actually applied.

In the introduction, the outputs of the Codemeta project are enumerated as the vocabulary and the crosswalks
with other vocabularies/schemas, repositories, registry and archives. I think it would be useful for
the reader if the paper also highlighted the software tools produced by the Codemeta project (https://codemeta.github.io/tools/).

When comparing the standardization processes of Codemeta and ISO, I think it is important to highlight that Codemeta follows an open process
and it is an open standard, while ISO is not and access to the information about ISO19115 is behind a paywall.

According to PeerJ policy (https://peerj.com/about/policies-and-procedures/#data-materials-sharing), which I commend, "Data should be provided in an appropriate,
machine-readable format. Note: formats such as PDF, Powerpoint, and images of tables etc. are not considered suitable for raw data sharing." Thus,
at a minimum, the mapping tables should be provided as CSV files. Ideally, it should be included in the Codemeta crosswalks (see comments about this later),
but otherwise, I suggest to include the mapping files in a repository such as Zenodo, so that they are versioned and given a DOI for their citation.

Experimental design

The mapping between the Codemeta vocabulary and the ISO19115 standard is presented in a very comprehensive way, with discussions
about the different challenges.

The goal of performing the mapping in this manuscript is presented as "facilitating the creation of codemeta-compliant
descriptions of software that is documented using the ISO standards" (from the introduction) and the mappings was
created "in order for ISO metadata creators and users to take advantage of the codemeta recommendations" (from the abstract).

To make this research question relevance more clear, it would be useful to see what are the use cases for ISO metadata creators to
convert to codemeta. What would be the scenarios of application? Also, if this is the objective, shouldn't the implementation
of such conversion be important and provided as one of the outputs (as one of the previous reviewers pointed out)?
As indicated above, I strongly recommend at a minimum that the mapping tables are made available in a machine-readable format.

Additionally, it is not clear to me why there are difficulties for this particular mapping to appear as a crosswalk file in the
codemeta repository (https://github.com/codemeta/codemeta/tree/master/crosswalks). In the conclusion, the author indicates
"Adding an ISO crosswalk to the codemeta framework was the original goal of this work, but the differences
identified and described here are significant enough to make that impossible." It seems to me that some of the challenges
presented in this paper coincide with the challenges of mapping other schemas to Codemeta. Checking for example the
mapping with DataCite(https://github.com/codemeta/codemeta/blob/master/crosswalks/DataCite.csv) and
considering that DataCite metadata is defined by an XML schema (http://schema.datacite.org/meta/kernel-4.1/), as it is
ISO19115, with some common characteristics with ISO19115 (e.g. relies on soft types rather than hard types),
it is not evident to me why this case is particularly difficult.

The conclusions section states that "the purpose [of the mapping] is different than the primary purpose of the crosswalks proposed
on the codemeta project site which is to facilitate automated translation between different JSON-LD vocabularies and RDF". Again,
this is not evident to me, as e.g. DataCite is based on an XML schema. I am aware of the tool that converts DataCite metadata into
schema.org-based JSON-LD, but the DataCite mapping (as linked above) is done based on the XML schema.

The analysis of the 'Dialect coverage and scope' is also useful for the Codemeta community. Is Figure 1 produced
based on the Codemedata full crosswalk table? Could the process be made available so that the figure could be updated
when crosswalks are revised or new crosswalks added?

The simplified XPath notation including only the role names and no XML-specific information is helpful to improve the readability
of the tables. However, given that there is no current actionable conversion (e.g. XSLT) of crosswalk in this paper,
it would be useful to provide the full XPath expressions as an additional table.

When describing 'Hard types and soft types', the author mentions RDF as an example for 'hard typing', i.e. where specific names
are required as names alone can be used to distinguish between items. However, this is not an RDF model constraint but rather
an implementation issue. In RDF, it is possible to use qualified relationships to work around the limitations of
binary predicates (see http://patterns.dataincubator.org/book/qualified-relation.html). Some of the conclusions on this respect
might need to be revised.

Table 2 presents the 'hard types' and associated ISO19115-1 codelist values grouped by item (e.g Dates, People and Organizations, etc).
Having the hard types and codelists lumped together doesn't allow to see the actual mappings. I suggest modifying this table
to provide more details, making clear that, for instance, Codemeta.datePublished maps to a date having CI_DateTypeCode.publication, while
on the other hand, there is no equivalent in Codemeta for CI_DateTypeCode.validityBegins and CI_DateTypeCode.validityExpires.

In the mapping tables, it seems that when there are no equivalent ISO 19115 terms, then the schema.org elements were not included. For example, in Table 6
representing the mapping from https://schema.org/Thing to ISO 19115, properties such as https://schema.org/alternateName or https://schema.org/image are not considered.
If not relevant, should the mapping table indicate this?

It would be helpful for the tables to include links to the properties information on the web. Please, also
note that some of the links in the Word document weren't converted as links in the PDF.

When referring to the schema.org entities, I suggest to present them without the whole path from Thing to the entity (i.e. schema:SoftwareApplication
rather than schema:Thing.CreativeWork.SoftwareApplication). If the hierarchy needs to be clarified, it can be done on the text rather than
when referring to the entity, as the schema.org namespace is flat and the presentation is otherwise confusing.

Validity of the findings

As indicated by one of the previous reviewers, the best way to validate the mapping would be to include it in the Codemeta crosswalks
(https://codemeta.github.io/crosswalk/). In the previous comments, I indicated that it is not clear why this seems to
be 'impossible' for ISO19115. Otherwise, it would be great to see ISO 19115-1 standard in the full crosswalk table
(https://github.com/codemeta/codemeta/blob/master/crosswalk.csv). Anyway, as a minimum, the mapping tables should be made available
in a machine-readable form.

If not the full code for the conversion, it would also be useful to have example instance data for ISO19115 and its Codemeta equivalent.

Given that Codemeta relies on the schema.org vocabulary, I wonder if the author considered the mappings developed by the W3C Spatial
Data on the Web Working Group: https://www.w3.org/2015/spatial/wiki/ISO_19115_-_DCAT_-_Schema.org_mapping ?

The conclusions provide a good summary of the challenges. I would like to see some discussion on how an RDF representation relying
on qualified relations would modify the interpretation of the mapping w.r.t. hard/soft types.

---

## Round 0.3 · Minor Revisions

Dear author,

Thanks again for having taken care of the reviewers' comments in you new revision. I've read carefully your answers in the rebuttal, and the modifications provided in this revision are convincing. However, there is still a minor – but really important, though – thing that should be fixed so as to have the article ready for publication, which still concerns one of the two comments I've presented in the past decision letter.

I totally understand that the development of an XSLT would be a quite huge and extensive work to do that is out-of-scope for this article (though it could also be very beneficial). However, I strongly feel that there is still an important contribution which is missing in the article, i.e. the description of the process you have followed for creating and validating the mapping.

I do believe, as you wrote, that the methodology used to develop crosswalks between metadata dialects is quite similar. But the fact that this methodology (that you have used as well according to your rebuttal) has not been described before in a way that is scientifically reproducible or reusable is exactly my point here. Thus, what I'm asking you is to provide, in a precise section of the article and according to your own experience with this work, a clear description of the methodology you have used for creating and then validating the proposed mapping - since I am pretty sure you have adopted a precise strategy for such development that deserves to be shared. This is a clear scientific contribution of your work that should not be implicitly mentioned, but rather it deserves a proper description in order to make your work more robust and reproducible, and so as to provide clear guidelines that anyone can follow for addressing the development of new crosswalks.

I'm really looking forward to reading this new addition to your work, so as to finally go through the publication process.

Have a nice day :-)

S.

---

## Round 0.4 · accepted · Accept

Dear author,

Thanks again for having addressed the issue raised in the previous round. I'm now happy with the current status of the paper, and we can proceed to publish it in PeerJ CS.

Have a nice day :-)

S.